# Learning to Control Self-Assembling Morphologies: A Study of Generalization via Modularity

**Deepak Pathak**[*]
UC Berkeley

**Chris Lu**[*]
UC Berkeley

**Trevor Darrell**
UC Berkeley

**Phillip Isola**
MIT

**Alexei A. Efros**
UC Berkeley

## Abstract

Contemporary sensorimotor learning approaches typically start with an existing complex agent (e.g., a robotic arm), which they learn to control. In contrast, this paper investigates a modular co-evolution strategy: a collection of primitive agents learns to dynamically self-assemble into composite bodies while also learning to coordinate their behavior to control these bodies. Each primitive agent consists of a limb with a motor attached at one end. Limbs may choose to link up to form collectives. When a limb initiates a link-up action, and there is another limb nearby, the latter is magnetically connected to the 'parent' limb's motor. This forms a new single agent, which may further link with other agents. In this way, complex morphologies can emerge, controlled by a policy whose architecture is in explicit correspondence with the morphology. We evaluate the performance of these *dynamic* and *modular* agents in simulated environments. We demonstrate better generalization to test-time changes both in the environment, as well as in the structure of the agent, compared to static and monolithic baselines. Project video and code are available at https://pathak22.github.io/modular-assemblies/.

## 1   Introduction

Possibly the single most pivotal event in the history of evolution was the point when single-celled organisms switched from always competing with each other for resources to sometimes cooperating, first by forming colonies, and later by merging into multicellular organisms [1]. These modular self-assemblies were successful because they combined the high adaptability of single-celled organisms while making it possible for vastly more complex behaviors to emerge. Indeed, one could argue that it is this modular design that allowed the multicellular organisms to successfully adapt, increase in complexity, and generalize to the constantly changing environment of prehistoric Earth. Like many researchers before us [13, 20, 23, 31, 32], we are inspired by the biology of multicellular evolution as a model for emergent complexity in artificial agents. Unlike most previous work, however, we are primarily focused on modularity as a way of improving adaptability and generalization to *novel test-time scenarios*.

In this paper, we present a study of modular self-assemblies of primitive agents — "limbs" — which can link up to solve a task. Limbs have the option to bind together by a magnet that connects their morphologies within the magnetic range (Figure 1), and when they do so, they pass messages and share rewards. Each limb comes with a simple neural net that controls the torque applied to its joints. Linking and unlinking are treated as dynamic actions so that the limb assembly can change shape within an episode. A similar setup has previously been explored in robotics as "self-reconfiguring modular robots" [21]. However, unlike prior work on such robots, where the control policies are hand-defined, we show how to *learn* the policies and study the generalization properties that emerge.

---

[*]Equal contribution.

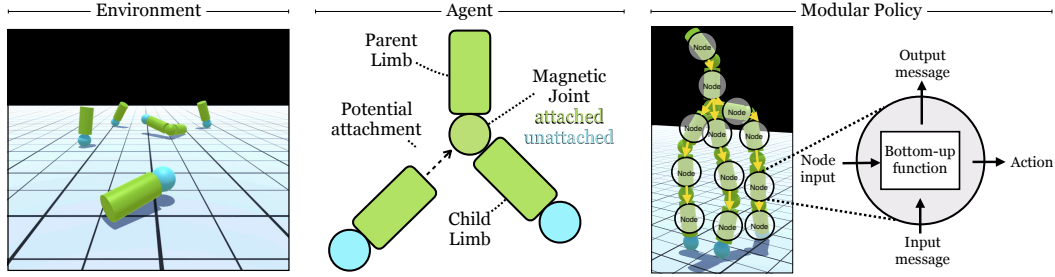

Figure 1: This work investigates the joint learning of control and morphology in self-assembling agents. Several primitive agents, containing a cylindrical body with a configurable motor, are dropped in a simulated environment (left). These primitive agents can self-assemble into collectives using magnetic joints (middle). The policy of the self-assembled agent is represented via proposed dynamic graph networks (DGN) with shared parameters (modular) across each limb (right).

Our self-assembled agent can be represented as a graph of primitive limbs. Limbs pass messages to their neighbors in this graph in order to coordinate behavior. All limbs have a common policy network with shared parameters, i.e., a modular policy which takes the messages from adjacent limbs as input and outputs torque to rotate the limb in addition to the linking/unlinking action. We call the aggregate neural network a "Dynamic Graph Network" (DGN) since it is a graph neural network [17] that can dynamically change topology as a function of its own outputs.

We test our dynamic limb assemblies on two separate tasks: standing up and locomotion. We are particularly interested in assessing how well can the assemblies generalize to novel testing conditions, not seen at training, compared to static and monolithic baselines. We evaluate test-time changes to both the environment (changing terrain geometry, environmental conditions), as well as the agent structure itself (changing the number of available limbs). We show that the dynamic self-assembles are better able to generalize to these changes than the baselines. For example, we find that a single modular policy is able to control multiple possible morphologies, even those not seen during training, e.g., a 6-limb policy, trained to build a 6-limb tower, can be applied at test time on 3 or 12 limbs, and still able to perform the task.

The main contributions of this paper are:

- Train primitive agents that self-assemble into complex morphologies to jointly solve control tasks.
- Formulate morphological search as a reinforcement learning (RL) problem, where linking and unlinking are treated as actions.
- Represent policy via modular dynamic graph network (DGN) whose topology matches the agent's physical structure.
- Demonstrate that self-assembling agents with dynamic morphology both train and generalize better than fixed-morphology baselines.

## 2   Environment and Agents

Investigating the co-evolution of control (i.e., *software*) and morphology (i.e., *hardware*) is not supported within standard benchmark environments typically used for sensorimotor control, requiring us to create our own. We opted for a minimalist design for our agents, the environment, and the reward structure, which is crucial to ensuring that the emergence of limb assemblies with complex morphologies is not forced, but happens naturally.

**Environment Structure**   Our environment contains an arena where a collection of primitive agent limbs can self-assemble to perform control tasks. This arena is a ground surface equipped with gravity and friction. The arena can be procedurally changed to generate a variety of novel terrains by changing the height of each tile on the ground (see Figure 2). To evaluate the generalization properties of our agents, we generate a series of novels terrains. This includes generating bumpy terrain by randomizing the height of nearby tiles, stairs by incrementally increasing the height of each row of tiles, hurdles by changing the height of each row of tiles, gaps by removing alternating rows of tiles, etc. Some variations also include putting the arena 'underwater', which basically amounts to

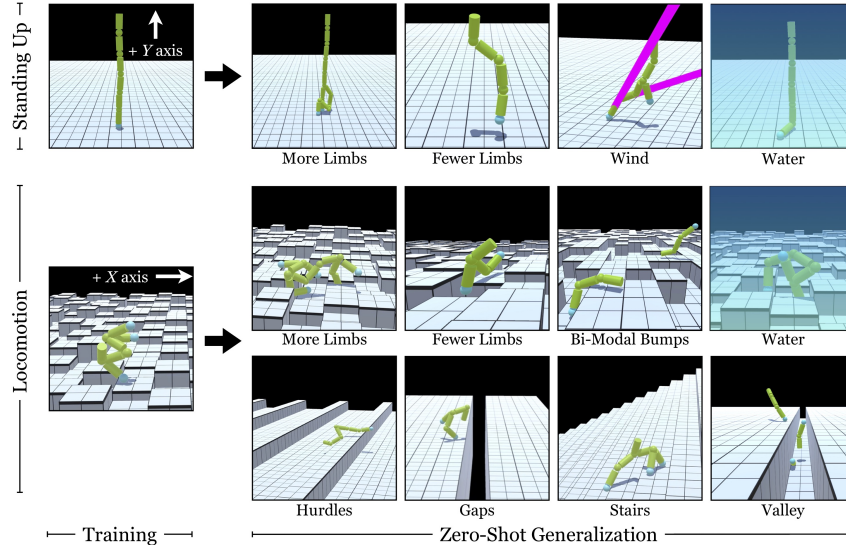

Figure 2: We illustrate our dynamic agents in two environments / tasks: standing up and locomotion. For each of these, we generate several new environment for evaluating generalization. Refer to project video at https://pathak22.github.io/modular-assemblies/ for better understanding of tasks.

increased drag (i.e., buoyancy). During training, we start our environment with a set of six primitive limbs on the ground, which can assemble to form collectives to perform complex tasks.

**Agent Structure**    All limbs share the same structure: a cylindrical body with a configurable motor on one end and the other end is free. The free end of the limb can link up with the motor-end of the other limb, and then the motor acts as a joint between two limbs with three degrees of rotation. Hence, one can refer to the motor-end of the cylindrical limb as a *parent-end* and the free end as a *child-end*. Multiple limbs can attach their child-end to the parent-end of another limb, as shown in Figure 1, to allow for complex graph morphologies to emerge. The limb at the parent-end controls the torques of the joint. The unlinking action can be easily implemented by detaching two limbs, but the linking action has to deal with the ambiguity of which limb to connect to (if at all). To resolve this, we implement the linking action by attaching the closest limb within a small radius around the parent-node. The attachment mechanism is driven by a magnet inside the parent node, which forces the closest child-limb within the magnetic range node to get docked onto itself if the parent signals to connect. The range of the magnetic force is approximately 1.33 times the length of a limb. If no other limb is present within the magnetic range, the linking action has no effect.

The primitive limbs are dropped in an environment to solve a given control task jointly. One key component of the self-assembling agent setup that makes it different from typical multi-agent scenarios [28] is that if some agents assemble to form a collective, the resulting morphology becomes a new *single agent* and all limbs within the morphology maximize a joint reward function. The output action space of each primitive agent contains the continuous torque values that are to be applied to the motor connected to the agent, and are denoted by $\{\tau_\alpha, \tau_\beta, \tau_\gamma\}$ for three degrees of rotation. The torque on parent and child limbs differs with respect to their configuration. The center of rotation of applied torque is the center of mass of the limb, and the orientation of axes is aligned with the limb's rotation. The torque applied by a limb is between the world and the limb itself and, hence, each limb directly only experiences the torque it exerts on itself. However, when it is connected to other limbs, its torque can affect its neighbors due to physical joints.

In addition to the torque controls, each limb can decide to attach another link at its parent-end or decide to unlink its child-end if already connected to other limbs. The linking and unlinking decisions are binary. This complementary role assignment of child and parent ends, i.e., a parent can only link, and a child can only unlink, makes it possible to decentralize the control across limbs in self-assembly.

**Sensory Inputs**    In our self-assembling setup, each agent limb only has access to its *local* sensory information and does not get any global information about other limbs of the morphology. The

sensory input of each agent includes its own dynamics, i.e., the location of the limb in 3-D euclidean coordinates, its velocity, angular rotation, and angular velocity. In order for the limb to combine with other limbs, it also gets access to the relative location of the nearest agent it can join with. Each end of the limb also has a trinary touch sensor to detect whether the end of the cylinder is touching 1) the floor, 2) another limb, or 3) nothing. Additionally, we also provide our limbs with a point depth sensor that captures the surface height on a $9 \times 9$ grid around the projection of the center of the limb on the surface. The surface height of a grid point is the vertical max-height (along Y-axis) of the surface of the tile at that point. This sensor is analogous to a simple camera and allows the limb to perceive its environment conditions.

One essential requirement to operationalize this setup is an efficient simulator to allow simultaneous simulation of several of these primitive limbs. We implement our environments in the Unity ML [11] framework, which is one of the dominant platforms for designing realistic games. The reason for picking Unity over other physics engine is to be able to simulate lots of limbs together efficiently. However, we keep the details like contact forces, control frequency, etc. quite similar to those in Mujoco gym environments. For computational reasons, we do not allow the emergence of cycles in the self-assembling agents by not allowing the limbs to link up with already attached limbs within the same morphology, although, our setup is easily extensible to general graphs. We now discuss the learning formulation for controlling our modular self-assemblies.

# 3 Learning to Control Self-Assemblies

Consider a set of primitive limbs indexed by $i$ in $\{1, 2, \ldots, n\}$ dropped in an environment arena $\mathcal{E}$ to perform a continuous control task. If needed, these limbs can assemble to form complex collectives in order to improve their performance on the task. The task is represented by a reward function $r_t$ and the goal of the limbs is to maximize the discounted sum of rewards over time $t$. If some limbs assemble into a collective, the resulting morphology effectively becomes a single agent with a combined policy to maximize the combined reward of the connected limbs. Further, the reward of an assembled morphology is a function of the whole morphology and not the individual agent limbs. For instance, in the task of learning to stand up, the reward is the height of the individual limbs if they are separate, but is the height of the whole morphology if those limbs have assembled into a collective.

## 3.1 Co-evolution: Linking/Unlinking as an Action

To learn a modular controller policy that could generalize to novel setups, our agents must learn the controller jointly as the morphology evolves over time. The limbs should simultaneously decide which torques to apply to their respective motors while taking into account the connected morphology. Our hypothesis is that if a controller policy could learn in a modular fashion over iterations of increasingly sophisticated morphologies (see Figure 3), it could learn to be robust and generalizable to diverse situations. So, how can we optimize control and morphology under a common end-to-end framework?

We propose to treat the decision of linking and unlinking as additional actions of our primitive limbs. The total action space $a_t$ at each iteration $t$ can be denoted as $\{\tau_\alpha, \tau_\beta, \tau_\gamma, \sigma_{link}, \sigma_{unlink}\}$ where $\tau_*$ denote the raw *continuous* torque values to be applied at the motor and $\sigma_*$ denote the *binary* actions whether to connect another limb at the parent-end or disconnect the child-end from the other already attached limb. This simple view of morphological evolution allows us to use ideas from RL [22].

## 3.2 Modularity: Self-Assembly as a Graph of Limbs

The integration of control and morphology in a common framework is only the first step. The key question is how to model this controller policy such that it is modular and reuses information across generations of morphologies. Let $a_t^i$ be the action space and $s_t^i$ be the local sensory input-space of the agent $i$. One naive approach to maximizing the reward is to simply combine the states of the limbs into the input-space, and output all the actions jointly using a single network. Formally, the policy is simply $\vec{a}_t = [a_t^0, a_t^1 \ldots a_t^n] = \Pi(s_t^0, s_t^0 \ldots, s_t^n)$. This interprets the self-assemblies as a single monolithic agent, ignoring the graphical structure. This is the current approach to solve many control problems, e.g., Mujoco humanoid [4], where the policy $\Pi$ is trained to maximize the sum of rewards using RL.

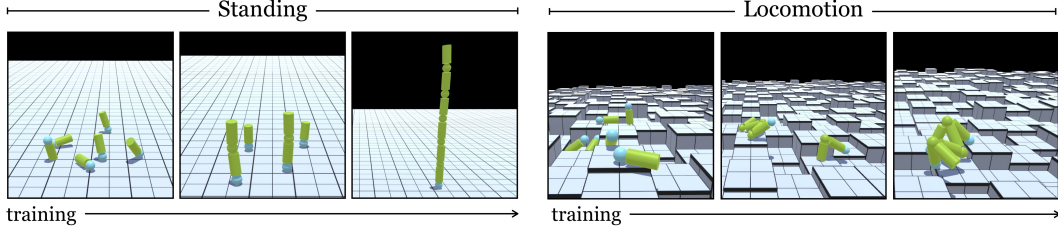

Figure 3: Co-evolution of Morphology w/ Control during Training: The gradual co-evolution of controller as well as the morphology of self-assembling agents over the course of training for the task of Standing Up (left) and Locomotion (right).

In this work, we represent the agent's policy via a graph neural network [17] in such a way that it explicitly corresponds to the morphology of the agent. Consider a collection of primitive limbs as graph $G$, where each node is a limb $i$. Two limbs being physically connected by a joint is analogous to having an edge in the graph. As discussed in Section 2, each limb has two endpoints, a *parent-end* and a *child-end*. At a joint, the limb which connects via its parent-end acts as a parent-node in the corresponding edge, and the other limbs, which connect via their child-ends, are child-nodes. The parent-node (i.e., the agent with the parent-end) controls the torque of the edge (i.e., the joint motor).

### 3.3 Dynamic Graph Networks (DGN)

Each primitive limb node $i$ has a policy controller of its own, which is represented by a neural network $\pi_\theta^i$ and receives a corresponding reward $r_t^i$ for each time step $t$. We represent the policy of the self-assembled agent by the aggregated neural network that is connected in the same graphical manner as the physical morphology. The edge connectivity of the graph is represented in the overall graph policy by passing messages that flow from each limb to the other limbs physically connected to it via a joint. The parameters $\theta$ are shared across each primitive limbs allowing the overall policy of the graph to be modular with respect to each node. However, recall that the agent morphologies are dynamic, i.e., the connectivity of the limbs changes based on policy outputs. This changes the edge connectivity of the corresponding graph network at every timestep, depending on the actions chosen by each limb's policy network in the previous timestep. Hence, we call this aggregate neural net a *Dynamic Graph Network (DGN)* since it is a graph neural network that can dynamically change topology as a function of its own outputs in the previous iteration.

**DGN Optimization**    A typical rollout of our self-assembling agents during an episode of training contains a sequence of torques $\tau_t^i$ and the linking actions $\sigma_t^i$ for each limb at each timestep $t$. The policy parameters $\theta$ are optimized to jointly maximize the reward for each limb:

$$\max_\theta \sum_{i=\{1,2...,n\}} \mathbb{E}_{\vec{a}^i \sim \pi_\theta^i}[\Sigma_t r_t^i] \tag{1}$$

We optimize this objective via policy gradients, in particular, PPO [19]. DGN pseudo-code (as well as source code) and all training implementation details and are in Section 1.1,1.4 of the supplementary.

**DGN Connectivity**    The topology is captured in the DGN by passing messages through the edges between individual network nodes. These messages are *learned* vectors passed from one limb to its connected neighbors. Since the parameters of these limb networks are shared across each node, these messages can be seen as context information that may inform the policy of its role in the corresponding connected component of the graph. Furthermore, as discussed in the previous section, each limb only receives its own *local* sensory information (e.g., its touch, depth sensor, etc.) and, hence, it can only get to know about far-away limbs states by *learning to pass* meaningful messages.

*(a) Message passing:* Since our agents have no cycles, the aggregated flow through the whole morphological graph can be encapsulated by passing messages in topological order. However, when the graph contains cycles, this idea can be easily extended by performing message-passing iteratively through the cycle until convergence, similar to loopy-belief-propagation in Bayesian graphs [14]. In this paper, messages are passed from leaf nodes to root, i.e., each agent gets information from its children. Instead of defining $\pi_\theta^i$ to be just as a function of state $s_t^i$, we pass each limb's policy

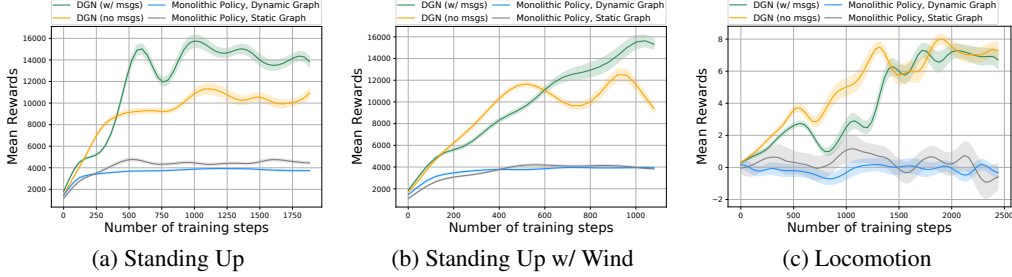

Figure 4: Training self-assembling agents: We show the performance of different methods for joint training of control and morphology for three tasks: learning to stand up (left), standing up in the presence of wind (center) and locomotion in bumpy terrain (right). These policies generalize to novel scenarios as shown in the tables.

network information about its children nodes. We redefine $\pi_\theta^i$ as $\pi_\theta^i : [s_t^i, m_t^{C_i}] \rightarrow [a_t^i, m_t^i]$ where $m_t^i$ is the output message of policy that goes into the parent limb and $m_t^{C_i}$ is the aggregated input messages from all the children nodes, i.e, $m_t^{C_i} = \sum_{c \in C_i} m_t^c$. If $i$ has no children (i.e, root), a vector of zeros is passed in $m_t^{C_i}$. Messages are passed recursively until the root node. An alternative way is to start from the root node and recursively pass until the messages reach the leaf nodes.

*(b) No message passing*: Note that for some environments or tasks, the context from the other nodes might not be a necessary requirement for effective control. In such scenarios, message passing might create extra overhead for training a DGN. Importantly, even with no messages, DGN still allows for coordination between limbs. This is similar to a typical cooperative multi-agent setup [28], where each limb makes its own decisions in response to the previous actions of the other agents. However, our setup differs in that our agents may physically join up, rather than just coordinating behavior.

## 4 Experiments

We test the co-evolution of morphology and control across two primary tasks where self-assembling agents learn to: (a) stand up, and (b) perform locomotion. Limbs start each episode disconnected and located just above the ground plane at random locations, as shown in Figure 3. In the absence of an edge, input messages are set to 0 and the output ones are ignored. Action space is continuous raw torque values. Across all the tasks, the number of limbs at training is kept fixed to 6. We take the model from each time step and evaluate it on 50 episode runs to plot mean and std-deviation confidence interval in training curves. At test, we report the mean reward across 50 episodes of 1200 environment steps. The main focus of our investigation is to evaluate if the emerged modular controller generalizes to novel morphologies and environments. Video is on the project website and implementation details are in Section 1.1 of the supplementary.

**Baselines** We further compare how well these dynamic morphologies perform in comparison to a learned monolithic policy for both dynamic and fixed morphologies. In particular, we compare to a (a) *Monolithic Policy, Dynamic Graph*: Baseline where agents are still dynamic and can self-assemble, but their controller is represented by a single monolithic policy that takes as input the combined state of all agents and outputs actions for each of them. (b) *Monolithic Policy, Fixed Graph*: Similar single monolithic policy as the previous baseline, but the morphology is hand-designed constructed from the limbs and kept fixed and static during training and test. This is analogous to a standard robotics "vanilla RL" setup in which morphology is predefined, and then a policy is learned to control it. We chose the fixed morphology to be a straight chain of 6-limbs in all the experiments. This linear-chain may be optimal for standing as tall as possible, but it is not necessarily optimal for *learning* to stand; the same would hold for locomotion. However, we confirmed that both standing and locomotion tasks are solvable with linear-chain morphology (shown in Figure 3 and video on the project website).

Although the monolithic policy is more expressive (complete state information of all limbs), it is also harder to train as we increase the number of limbs, because the observation and action spaces increase in dimensionality. Indeed, this is what we find in Figure 1 of supplementary: the monolithic policy can perform well on up to three limbs but does not reach the optimum on four to six limbs. In contrast, the DGN limb policy (shared between all limbs) has a fixed size observation and action space, independent of the number of limbs under control.

| Environment | DGN (ours) | Monolithic Policy | |
| --- | --- | --- | --- |
| | | (dynamic) | (fixed) |
| *Standing Up Task* | | | |
| *Training Environment* | | | |
| Standing Up | **17518** | 4104 | 5351 |
| *Zero-Shot Generalization* | | | |
| More (2x) Limbs | **19796** (113%) | n/a | n/a |
| Fewer (.5x) Limbs | **10839** (62%) | n/a | n/a |
| *Standing Up in the Wind Task* | | | |
| *Training Environment* | | | |
| Stand-Up in Wind | **18423** | 4176 | 4500 |
| *Zero-Shot Generalization* | | | |
| 2x Limbs + (S)Wind | **15351** (83%) | n/a | n/a |
| *Locomotion Task* | | | |
| *Training Environment* | | | |
| Locomotion | **8.71** | 0.96 | 2.96 |
| *Zero-Shot Generalization* | | | |
| More (2x) Limbs | **5.47** (63%) | n/a | n/a |
| Fewer (.5x) Limbs | **6.64** (76%) | n/a | n/a |

Table 1: Zero-Shot Generalization to Number of Limbs: Quantitative evaluation of the generalizability of the learned policies. For each method, we first pick the best performing model from the training and then evaluate it on each of the novel scenarios without further finetuning, i.e., in a zero-shot manner. We report the score attained by the self-assembling agent along with the percentage of training performance retained upon transfer in parenthesis. Higher is better.

| Environment | DGN (ours) | Monolithic Policy | |
| --- | --- | --- | --- |
| | | (dynamic) | (fixed) |
| *Standing Up Task* | | | |
| *Training Environment* | | | |
| Standing Up | **17518** | 4104 | 5351 |
| *Zero-Shot Generalization* | | | |
| Water + 2x Limbs | **16871** (96%) | n/a | n/a |
| Winds | **16803** (96%) | 3923 (96%) | 4531 (85%) |
| Strong Winds | **15853** (90%) | 3937 (96%) | 4961 (93%) |
| *Standing Up in the Wind Task* | | | |
| *Training Environment* | | | |
| Stand-Up in Wind | **18423** | 4176 | 4500 |
| *Zero-Shot Generalization* | | | |
| (S)trong Wind | **17384** (94%) | 4010 (96%) | 4507 (100%) |
| Water+2x+SWd | **17068** (93%) | n/a | n/a |
| *Locomotion Task* | | | |
| *Training Environment* | | | |
| Locomotion | **8.71** | 0.96 | 2.96 |
| *Zero-Shot Generalization* | | | |
| Water + 2x Limbs | **6.57** (75%) | n/a | n/a |
| Hurdles | **6.39** (73%) | -0.77 (-79%) | -3.12 (-104%) |
| Gaps in Terrain | **3.25** (37%) | -0.32 (-33%) | 2.09 (71%) |
| Bi-modal Bumps | **6.62** (76%) | -0.56 (-57%) | -0.44 (-14%) |
| Stairs | **6.6** (76%) | -8.8 (-912%) | -3.65 (-122%) |
| Inside Valley | **5.29** (61%) | 0.47 (48%) | -1.35 (-45%) |

Table 2: Zero-Shot Generalization to Novel Environments: The best performing model from the training is evaluated on each of the novel scenarios without any further finetuning. The score attained by the self-assembling agent is reported along with the percentage of training performance retained upon transfer in parenthesis. Higher value is better.

## 4.1 Learning to Self-Assemble

We first validate if it is possible to train the self-assembling policy end-to-end via Dynamic Graph Networks. Below, we discuss our environments and compare the training efficiency of each method.

**Standing Up Task**   In this task, each agent's objective is to maximize the height of the highest point in its morphology. Limbs have an incentive to self-assemble because the potential reward would scale with the number of limbs if the self-assembled agent can control them. The training process begins with six-limbs falling on the ground randomly, as shown in Figure 3. These limbs act independently in the beginning but gradually learn to self-assemble as training proceeds. Figure 4a compares the training efficiency and performance of different methods during training. We found that our DGN policy variants perform significantly better than the monolithic policies for standing up the task.

**Standing Up in the Wind Task**   Same as the previous task, except with the addition of 'wind', which we operationalize as random forces applied to random points of each limb at random times, see Figure 2(Wind). Figure 4b shows the superior performance of DGN compared to the baselines.

**Locomotion Task**   The reward function for locomotion is defined as the distance covered by the agent along $X$-axis. The training is performed on a bumpy terrain shown in Figure 2. The training performance in Figure 4c shows that DGN variants outperform the monolithic baselines.

As shown in Figure 4, training our DGN algorithm with message passing either seems to perform better or similar to the one without message passing. In particular, message passing is significantly helpful where long-term reasoning is needed across limbs, for instance, messages help in standing up the task because there is only one morphological structure to do well (i.e., linear tower). In locomotion, it is possible to do well with a large variety of morphologies, and thus both DGN variants reach similar performance. We now show results using DGN w/ msgs as our primary approach.

Furthermore, we trained a modular DGN policy for static morphology to see whether it is the modularity of policy (software), or modularity of the physical morphology of agent (hardware), that allows the agent to work well. These results are shown in Figure 3 of the supplementary. The performance is significantly better than 'monolithic policy, static graph' but worse than our final self-assembling DGN, which suggests that both modularity of software, as well as hardware, are necessary for successful training and generalization.

## 4.2 Zero-Shot Generalization to Number of Limbs

We investigate if our trained policy generalizes to changes in the number of limbs. We pick the best model from training and evaluate it without any finetuning at test-time, i.e., zero-shot generalization.

**Standing Up Task** We train the policy with 6 limbs and test with 12 and 4 limbs. As shown in Table 1, despite changes in the number of limbs, DGN is able to retain similar performance w/o any finetuning. We also show this variation in the max-performance of the DGN agent as the number of limbs changes in Figure 2 of the supplementary material. The co-evolution of morphology jointly with the controller allows the modular policy to experience increasingly complex morphological structures. We hypothesize that this morphological curriculum at training makes the agent more robust at test-time.

Note that we can not generalize *Monolithic policy* baselines to scenarios with more or fewer limbs because they can't accommodate different action and state-space dimensions from training; it has to be retrained. Hence, we made a comparison to DGN by retraining baseline on Standing task: DGN is trained on 6 limbs and tested on 4 limbs w/o any finetuning, while baseline is trained both times. DGN achieves 17518 (6limbs - train), 10839 (4limbs - test) scores, while baseline achieves 5351 (6limbs - train), 7356 (4limbs - train). Even without any training on 4 limbs, DGN outperforms baseline because it is difficult to train monolithic policy with large action space (Figure 1 in Appendix).

**Standing Up in the Wind Task** Similarly, we evaluate the agent policy trained for standing up task in winds with 6 limbs to 12 limbs. Table 1 shows that the DGN performs significantly better than a monolithic policy at training and able to retain most of its performance even with twice the limbs.

**Locomotion Task** We also evaluate the generalization of locomotion policies trained with 6 limbs to 12 and 4 limbs. As shown in Table 1, DGN not only achieves good performance at training but is also able to retain most of its performance.

## 4.3 Zero-Shot Generalization to Novel Environments

We now evaluate the performance of our modular agents in novel terrains by creating several different scenarios by varying environment conditions (described in Section 2) to test zero-shot generalization.

**Standing Up Task** We test our trained policy without any further finetuning in environments with increased drag (i.e., 'under water'), and adding varying strength of random push-n-pulls (i.e. , 'wind'). Table 2 shows that DGN seems to generalize better than monolithic policies. We believe that this generalization is a result of both the learning being modular as well as the fact that limbs learned to assemble in physical conditions (e.g., forces like gravity) with gradually growing morphologies. Such forces with changing morphology are similar to setup with varying forces acting on fixed morphology resulting in robustness to external interventions like winds. As discussed in previous subsection, generalization provides clear advantages across number of limbs (Table 1), but in Table 2, baseline generalization performance for standing-up task is also more than $90\%$. This suggests that the trainability of these agents correlates with generalization in standing task. Hence, one could argue that the potential benefit of our method is that it trains better, which partially explains its high performance at test time generalization. Although, in the locomotion experiments, the generalization gap (the difference between training and test performance) is substantially lower for our method compared to the baselines, which reaffirms that modularity improves trainability as well as generalization.

**Standing Up in the Wind Task** Similarly, the policies trained with winds are able to generalize to scenarios with either stronger winds or winds inside water.

**Locomotion Task**   We generate several novel scenarios for evaluating locomotion: with water, a terrain with hurdles of a certain height, a terrain with gaps between platforms, a bumpy terrain with a bi-modal distribution of bump heights, stairs, and an environment with a valley surrounded by walls on both sides (see Figure 2). These variations are generated procedurally. The modular policies learned by DGN tend to generalize better than the monolithic agent policies, as shown in Table 2.

This generalization could be explained by the incrementally increasing complexity of self-assembling agents at training. For instance, the training begins with all limbs separate, which gradually form a group of two, three, and so on, until the training converges. Since the policy is *modular with shared parameters across limbs*, the training of smaller size assemblies with small bumps would, in turn, prepare the large assemblies for performing locomotion through higher hurdles, stairs, etc. at test. Furthermore, the training terrain has a finite length, which makes the self-assemblies launch themselves forward as far as possible upon reaching the boundary to maximize the distance along X-axis. This behavior helps the limbs generalize to environments like gaps or valley where they end up on the next terrain upon jumping and continue to perform locomotion.

## 5   Related Work

**Morphologenesis & self-reconfiguring modular robots**   The idea of modular and self-assembling agents goes back at least to Von Neumman's *Theory of Self-Reproducing Automata* [24]. In robotics, such systems have been termed "self-reconfiguring modular robots" [13, 21]. There has been a lot of work in modular robotics to design real hardware robotic modules that can be docked together to form complex robotic morphologies [5, 8, 15, 29, 31]. Alternatives to optimize agent morphologies include genetic algorithms that search over a generative grammar [20] and energy-based minimization to directly optimizing controllers [6, 25]. Schaff et al. [18] improves the design of individual limbs keeping the morphology fixed. We approach morphogenesis from a learning perspective, in particular, deep RL, and study the resulting generalization properties. We achieve morphological co-evolution via *dynamic actions* (linking), which agents take during their lifetimes, whereas the past approaches treat morphology as an optimization target to be updated between generations or episodes. Since the physical morphology also defines the connectivity of the policy net, our proposed algorithm can also be viewed as performing a kind of neural architecture search [33] in physical agents.

**Graph neural networks**   Encoding graphical structures into neural networks [17] has been used for a large number of applications, including question answering [2], quantum chemistry [7], semi-supervised classification [12], and representation learning [30]. The works most similar to ours involve learning controllers [16, 27]. For example, Nervenet [27] represents individual limbs and joints as nodes in a graph and demonstrates multi-limb generalization. However, the morphologies on which Nervenet operates are not learned jointly with the policy and hand-defined to be compositional in nature. Others [3, 10] have shown that graph neural networks can also be applied to inference models as well as to planning. Prior graph neural network-based approaches deal with a static graph, which is defined by auxiliary information, e.g., language parser [2]. In contrast, we propose dynamic graph networks where the graph policy changes itself dynamically over the training.

**Concurrent Work**   Ha [9] use RL to improve limb design given fixed morphology. Wang et al. [26] gradually evolves the environment to improve the robustness of an agent. However, both the work assumes the topology of agent morphology to stay the same during train and test.

## 6   Discussion

Modeling intelligent agents as modular, self-assembling morphologies has long been a very appealing idea. The efforts to create practical systems to evolve artificial agents goes back at least two decades to the beautiful work of Karl Sims [20]. In this paper, we are revisiting these ideas using the contemporary machinery of deep networks and reinforcement learning. Examining the problem in the context of machine learning, rather than optimization, we are particularly interested in modularity as a key to generalization in terms of improving adaptability and robustness to novel environmental conditions. Poor generalization is the Achilles' heel of modern robotics research, and the hope is that this could be a promising direction in addressing this key issue. We demonstrated a number of promising experimental results, suggesting that modularity does indeed improve generalization in simulated agents. While these are just the initial steps, we believe that the proposed research direction is promising, and its exploration will be fruitful to the research community. To encourage follow-up work, we have publicly released all code, models, and environments on the project webpage.

## Acknowledgments

We would like to thank Igor Mordatch, Chris Atkeson, Abhinav Gupta, and the members of BAIR for fruitful discussions and comments. This work was supported in part by Berkeley DeepDrive and the Valrhona reinforcement learning fellowship. DP is supported by the Facebook graduate fellowship.

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
