[Supplementary Material]

# Supplementary Material for
# Learning to Control Self-Assembling Morphologies:
# A Study of Generalization via Modularity

**Deepak Pathak**[*]  **Chris Lu**[*]   **Trevor Darrell**   **Phillip Isola**   **Alexei A. Efros**
UC Berkeley      UC Berkeley      UC Berkeley          MIT              UC Berkeley

## 1 Appendix

### 1.1 Implementation and Training details

We use PPO as the underlying reinforcement learning method to optimize the joint DGN objective shown in Equation (1) in Section 3.3 (main paper). Each limb policy is represented by a 4-layered fully-connected neural network with ReLU non-linearities and trained with a learning rate of $3e - 4$, a discount factor of $0.995$, entropy coefficient of $0.01$, advantage parameter of $0.95$ and a batch size of $2048$. The messages are 32 length float vectors. The optimizer used to optimize PPO is RMS-Prop. Parameters are shared across network modules, and they all predict action at the same time. Each episode is 5000 steps long at training. Across all the tasks, the number of limbs at training is kept fixed to 6. Limbs start each episode disconnected and located just above the ground plane at random locations, as shown in Figure 3 in the main paper. In the absence of an edge, input messages are set to 0, and the output ones are ignored. Action space is continuous raw torque values. We take the model from each time step and evaluate it on 50 episodes to plot mean and standard-deviation (confidence intervals) in training curves. At the test time, we report the mean reward across 50 episode runs of 1200 environment steps.

### 1.2 Monolithic Policy w/ Static Morphology Basline vs. Number of Limbs

To verify whether the training of *Monolithic Policy w/ Static Graph* is working, we ran it on standing up and locomotion tasks across a varying numbers of limbs. We show in Figure 1 that the baseline performs well with fewer limbs, which suggests that the reason for failure in the 6-limbs case is indeed the morphology graph being fixed and not the implementation of this baseline.

(a) Standing Up                (b) Locomotion

Figure 1: The performance of *Monolithic Policy w/ Static Morphology* baseline as the number of limbs varies in the two tasks: standing up (left) and locomotion (right). This shows that the monolithic baseline works well with less (1-3 limbs), but fails with 6 limbs during training.

In contrast, our method can easily train for 6-limbs (where fixed graph fails), and the same model generalizes to other limbs as well. To show this ablation, we show how does the max-performance of

the agent varies as the number of limbs changes for an agent that was trained on 6-limbs. Figure 2 shows the change in max-performance after k steps as a function of the number of limbs in case of Standing Up task for our DGN (w/ msgs) model. A quantitative evaluation of this generalization is discussed in Table 1 in the main paper.

## 1.3 Performance of Modular DGN Policy on Static Morphology

We trained a modular DGN policy for static morphology to see whether it is the modularity of policy (software), or modularity of the physical morphology of agent (hardware), that allows the agent to work well. We augment this baseline to the plots from Figure 4 of the main paper. As shown in Figure 3, the performance is significantly better than 'monolithic policy, static graph' but worse than our final self-assembling DGN. This suggests that the modularity of software, as well as hardware, are necessary for successful training and generalization. Nevertheless, regardless of generalization properties, one of the main contribution of our work is showing how could dynamic agents be trained to self-assemble.

Figure 2: Performance of DGN (w/ msgs) agent trained with 6-limbs for Standing Up task with respect to change in # of limbs.

## 1.4 Pseudo Code of the DGN Algorithm

Notation is summarized in Algorithm 1, and the full pseudo-code is summarized in the following algorithm boxes: Algorithm 2 , and Algorithm 3. Full source code available at https://pathak22.github.io/modular-assemblies/.

(a) Standing Up
(b) Locomotion

Figure 3: The performance of different methods for joint training of control and morphology for three tasks with an additional baseline for training static morphology agent with a modular DGN policy: learning to stand up (left) and locomotion in bumpy terrain (right). The training plots are extension Figure 4 of the main paper.

---

**Algorithm 1:** Notation Summary (defined in Section 3.3)

1 **foreach** *node i* **do**
2 $\quad\mid\quad a_t^i, m_t^i = \pi_\theta^i(s_t^i, m_t^{C_i})$
3 **end**
4 **where**
5 $s_t^i$: observation state of agent limb i
6 $a_t^i$: action output of agent limb i: 3 torques, attach, detach
7 $m_t^{C_i}$: aggregated message from children nodes input to agent i (bottom-up-1)
8 $m_t^i$: output message that agent i passes to its parent (bottom-up-2)
9 $\theta$: $\theta_1, \theta_2$
10 messages are 32 length floating point vectors.

---

**Algorithm 2:** Pseudo-code: DGN w/ Message Passing

1   Initialize parameters $\theta_1, \theta_2$ randomly.
2   Initialize all message vectors $m_t^{C_i}, m_t^i$ to be zero
3   Represent graph connectivity $G$ as a list of edges
4   Note: In the beginning, all edges are zeros, i.e., non-existent
5   **foreach** *timestep* $t$ **do**
6      Each limb agent $i$ observes its own state vector $s_t^i$
7      **foreach** *agent* $i$ **do**
8         # Compute incoming child messages
9         $m_t^{C_i} = 0$
10        **foreach** *child node* $c$ *of agent* $i$ **do**
11           $m_t^{C_i} += m_t^c$
12        **end**
13        # Compute action and message to parent $p$ of agent $i$ in $G$
14        $a_t^i, m_t^i := \pi_\theta^i(s_t^i, m_t^{C_i})$
15        # Execute morphology change as per $a_t^i$
16        **if** $a_t^i[3] ==$ *attach* **then**
17           find closest agent $j$ within distance $d$ from agent $i$, otherwise $j$=NULL
18           add edge $(i, j)$ in $G$
19           also make physical joint between $(i, j)$
20        **end**
21        **if** $a_t^i[4] ==$ *detach* **then**
22           delete edge $(i,$ parent of $i)$ in $G$
23           also delete physical joint between $(i, j)$
24        **end**
25        # Execute torques from $a_t^i$
26        Apply torques $a_t^i[0], a_t^i[1], a_t^i[2]$
27      **end**
28      # Update graph and agent morphology
29      Find all connected components in $G$
30      **foreach** *connected component* $c$ **do**
31        **foreach** *agent* $i \in c$ **do**
32           reward $r_t^i =$ reward of $c$ (e.g. max height)
33        **end**
34      **end**
35   **end**
36   Update $\theta$ to maximize discounted reward using PPO as follows:
37   let $\vec{a}_t = [a_t^1, a_t^2 .. a_t^n]$
38      $\vec{s}_t = [s_t^1, s_t^2 .. s_t^n]$
39      $\hat{A}_t =$ advantage of discounted rewards, $r_t = \sum_{agent\, i} r_t^i$
40   PPO: $\max_\theta \mathbb{E}[\hat{A}_t \frac{\pi_\theta(\vec{a}_t|\vec{s}_t)}{\pi_{\theta_{old}}(\vec{a}_t|\vec{s}_t)} - \beta \mathrm{KL}(\pi_{\theta_{old}}(.|\vec{s}_t)\pi_\theta(.|\vec{s}_t))]$
41   Repeat until training converges

---

**Algorithm 3:** Pseudo-code: No-message DGN

1   Same as Algorithm-2 but hard-code incoming child and parent messages to be always 0, i.e.,
    $m_t^{C_i} = 0$ and $m_t^{p_i} = 0$ in each iteration.