[Reviews · NeurIPS 2019]

Reviewer 1



Summary: This paper explores control with modular limbs which can change dynamically change their configurations as a part of their action space. This paper defines a limb, which can connect itself to other limbs and exert torques. It is a basic building block from which the agents construct their bodies. This paper limits the morphologies to tree like structures but could be generalized. A single controller neural network is shared across all of these limbs and therefore decisions about their actions are done in a distributed fashion. These limbs communicate in 1 direction when they are connected with a setup similar to an RNN. This agents successfully learn to connect into bigger bodies and solve tasks such as standing and locomotion. As baseline the paper chooses an agents which can the fully observe and act on all of the links. The experiments are aimed at showing a better generalization to novel scenarios than the baseline. However every algorithm is only run once and details of the task are not clearly described. The baseline algorithm fails to learn properly with the given number of limbs, which is unexpected given other work in the field on high-dof control systems. The results also do prove that this approach generalizes well across number of limbs. Specifically during locomotion the algorithm learns a policy which is very robust with respect to loosing limbs. Overall the idea is novel, the paper is largely well-written, and the positive results are promising. In particular the zero-shot generalization to a different number of limbs is a compelling result, which is not possible for non-modular architectures. The main caveat is that the baselines failed on all 3 tasks, which suggest a poor choice of evaluation domains or lack of effort on baseline performance. Comments: The idea and approach presented in this paper is very nice. These agents are modular and learn reassemble themselves according to the context. Also the application of Dynamic Graph Networks is nice as they are well matched to the problem. The choice of baseline also seems appropriate as it well represents the currently used approaches. In the explanation however I found several parts which I would have prefered to be clearer: Action space: The linking and unlinking is described at several places within the paper but it is not clear to me how exactly it works and which limbs is responsible for what. My understanding is that: linking is initiated by the parent, unlinking by the child and torques are determined by the parent. Does that mean that each child node connected to the same parent experiences the same torque? What happens when a node does not have any children? In classical control tasks (e.g. cheetah, humanoid) such torques would not be possible but the video shows a controllable single limb bodies. My only interpretation is that the torque is between the limb and the world which is not mentioned anywhere though. Also there is no explanation what coordinate frames are these torques in (ie. body or world). It would also be good to include a description of the range of the magnetic force and torque ranges. In second 43 of the video we see 2 limbs flying directly up which cannot be achieved by torques alone. Is the magnetic range longer than 1 link? Observation space: I miss a more detailed description of the observations. I assume the position 3-dim cartesian, but is unclear what is the representation for the rotation and whether there were any convolution layers applied to the 9*9 height grid or how big the grid was. Communication: You describe DNN as a comunication structure. However later it is described that the information only flows in 1 direction and accumulates along the links. This setup is more similar to a RNN than a GNN. It is not clear whether more communication architectures were tried and what was the reason to choose this one. Also it is not clear how the links manage to find each other at the beginning. The information is only shared across connected bodies and so they can dynamically decide how to meet. In stand up case they can "agree" on specific x-y coordinates but it is not clear how it can work in the locomotion case. Baselines and evaluation: This is very similar to a multi-agent setup which can be unstable to optimize and often have big fluctuations in their performance based on the random seed. The experiments as they are described are only ran once for each algorithm and seed which makes it hard to judge how big the gaps in the performance really are. The modularity provides a very clear advantages in terms of generalizations across number of lines but at the top of table 2 all the generalizations fall to >90% of their baseline performance. This would suggest that the biggest benefit might be the trainability of these systems. It is also not clear why the 6 link baselines failed to learn. The reasoning presented in the paper and supplementary material is that the action space was too big but 6 links * 3 actions = 18 dimensional action space which is comparable to humanoids where these agents still work. It was also not clear to me what the water environment is as both buoyancy and drag are mentioned.

Reviewer 2



As the authors note, the idea of reasoning over both the design of an agent and its control policy is not new, with work in the robotics community dating back at least several decades. However, the large majority of recent work has focused exclusively on control in the context of a given design, with particular emphasis on policies learned via deep RL. Some recent methods have been proposed within the robotics and learning communities that revisit the idea of joint optimization, either via model-based trajectory optimization or model-free RL approaches. This paper continues in this direction by considering the problem of including the agent’s morphology as part of the design space, which poses interesting optimization challenges due the fact that the space is discrete. The paper describes a framework for jointly optimizing over the space of morphologies and designs using RL (which is better suited to the challenges posed by the discrete-continuous space). Modifications to the morphology (attachment and detachment by individual links) are treated as part of the action space, with a policy that is aware of the dynamic structure of the agent. As noted, the paper considers an interesting problem (joint optimization over a hybrid discrete-continuous design and control space) and describes an approach that seems sensible (essentially, treating the ability to connect/disconnect from an adjacent link as another action). I find this to be a valuable contribution. However, the paper leaves several open questions that make it difficult to draw conclusions about the effectiveness of this approach. The paper provides few details regarding the specific nature of the architecture and the training procedure. Hybrid policies such as the one proposed have been found to be difficult to train, requiring careful thought about how one trades off between attachment/detachment and learning the control policy for the current morphology. Did the authors find the architecture to be similarly sensitive to this scheduling? The training and evaluation consider environments that are not standard in the community. The need for a new environment is attributed to limitations of existing environments when it comes to modifications to the topology. This is unfortunate as this makes it difficult to judge the richness of the environment (e.g., are ground contact forces modeled? How about contact between limbs?). It is not clear why environments built around Mujoco or PyBullet (e.g., OpenAI Gym) would not allow for variations in the topology. The paper makes vague references to “messages” communicated between limbs, but the nature of these messages is not clear (as is their relationship to the claim that limbs aren’t aware of other limbs). The proposed framework is evaluated relative to baselines, however the baselines are weak. They include designs hand-crafted by the authors rather than design experts and the fact that the task is solvable with these designs isn’t sufficient to justify their choice. It would be more appropriate to consider a baseline that operated on random morphologies and one that employed Bayesian optimization (an evolutionary approach would be fitting). Given the relative small space of topologies, a random baseline would presumably perform fairly well. I was surprised not to see a qualitative evaluation of the learned morphologies as well as the learned motion policies for the different settings. The related work discussion is fairly thorough, with references to relevant decades-old work from the robotics community, which is refreshing. However, I would like to see a more thorough discussion relative to recent work on joint optimization of design and control and call into question the statement that work by Schaff et. al is “concurrent work” as it first appeared in January 2018. At today’s rate, 14+ months is hardly concurrent work. ADDITIONAL COMMENTS/QUESTIONS * It isn’t clear how much the controller on each link knows about the overall morphology. In lines 97-98, it is stated that each limb does not know about other limbs (though it has access to a sensor that indicates whether or not it is connected to another limb?), yet the controller on each limb reasons over the morphology. Aren’t these two things inconsistent. * About which axis is the surface height grid projected? * The distinction between using vs. not using messages is not clear as this would be a design decision at the outset. * The paper should report the standard deviation of the reward at test time. MINOR: There are a few grammatical errors (e.g., lines 81, )

Reviewer 3



The paper looked into an interesting problem of training self-assembling modularized robots and investigated the generalization of different training methods. Though there has been work in self-assembling robots or training modular robots with graph neural networks, training modular robots that can change its morphology through self-assembling is novel in my knowledge. The paper is well written and easy to follow. The experiments shown in this work consists of two tasks: stand up and locomotion, and the resulting policies are tested to generalize to different perturbations and morphologies. One question I have about the method is that it’s not completely clear to me how important the self-assembling part is to the generalization of the resulting policy? For example, if during training, one uses the graph neural networks representation and trains on a set of manually designed morphologies, is it also going to generalize to other morphologies? Another question about the implementation of the self-assembling robot. From the video it seems that the robot sometimes breaks apart into multiple parts. Are those cases all due to the individual agent producing the unlink action, or the robot might break in other conditions (like due to large external force, or being taken by another link). In general it might be interesting to visualize the actions of linking and unlinking during the rollout. Overall, I think this is an interesting paper with impressive results.

[Author Response · NeurIPS 2019]

We thank the reviewers for their feedback. The reviewers R1 and R3 suggested additional experiments. We are pleased
to report that we have completed those experiments. We report those results and address other concerns below. Due to
limited space, we couldn't answer all of the reviewers' clarification queries but promise to include in the final version.

**[R1] "why the 6 link baselines failed... comparable to humanoids where these agents still work"**: In Humanoid
environments (e.g. in OpenAI Gym), the range of each joint angle is carefully set with hand-designed limits using
domain knowledge (e.g., abdomen can't bend as much as knee) which makes it possible to train up to 17 DOF. In
contrast, our agents have no joint angle, or torque, limits which makes the task much harder. Hence, as shown in
Supplementary Figure 1, the monolithic baseline works until 4 limbs (i.e., 12 DOF), but fails to scale beyond that.

**[R1] "...child node connected to the same parent experiences the same torque? What happens when a node does
not have any children? coordinate frames are these torques?"**: The torque on parent/child differs with respect to
their location/configuration. The center of rotation of applied torque is the center of mass of the limb, and the axes
orientation are aligned with the limb's rotation. Hence, each limb directly only experiences the torque it exerts on itself.
However, when it is connected to other limbs, its torque can affect its neighbors physically.
Yes, in absence of children, torque is between the world and the limb itself. We will add it to the environment details.

**[R1] "range of magnetic force?"**: The range of the magnetic force is approximately 1.33 times the length of a limb.

**[R1] "...setup is more similar to a RNN than a GNN"**: We refer to it as graph because the topology is tree structured.
Standard RNNs don't have this topology and are usually applied in a linear sequence.

**[R1] "generalizations fall to >90%... benefit might be the trainability"**: This is
a good point. We will clarify that one benefit of our method is that it trains better,
which partially explains its high performance at test time. However, in the locomotion
experiments, we see that the generalization gap (the difference between training and
test performance) is substantially lower for our method compared to the baselines.

**[R1] plot "max-performance after k steps as a function of number of links"**: We
show this plot for standing task in Figure 1 for our DGN (w/ msgs) model.

Figure 1: DGN in standing task:
performance wrt # of limbs.

**[R1, R2] show "standard deviation of runs at test time"**: We trained two models per
experiment with 50 episodic evaluations for each model at every checkpoint time-step.
The numbers reported in table are mean performance scores. We couldn't report std-errors in Tables due to space
congestion, and will include them in the final version (1 extra page is allowed in camera ready).

**[R2] "surprised not to see qualitative evaluation of learned morphologies, policies"**: Qualitative results are pro-
vided in the supplementary as a video (`ProjectVideo.mp4`) for both of the training tasks and 8 evaluation scenarios,
together with baselines and narrated audio. In case R2 missed the video, we highly encourage them to check it out.

**[R2] "architecture, training procedure details.. important to replicate results"**: Supplementary material contains
(a) training details (Sec A.1), (b) pseudo-code algorithm box (Sec A.4), (c) full source code for reproducibility.

**[R2] "environments that are not standard... why not environments built around Mujoco"**: We did try Mujoco
briefly but found it too slow to simulate lots of individual *controllable* limbs in parallel. Hence, we switched to the
standard Unity ML-agents framework [Juliani et.al. 2018], which is a dominant platforms for designing realistic games
and is efficient. Details like contact forces, control frequency etc. are kept as similar to Mujoco as possible.

**[R2] "nature of these messages is not clear... how much the controller on each link knows about the overall
morphology"**: We describe how messages are passed in Lines 166-182. Briefly, they are 32-dim *learned* vectors passed
from one limb to its connected neighbors. Each limb only receives its own sensory data (e.g. its touch, depth sensor),
and can only get to know about far-away limbs states by *learning to pass* meaningful messages. An algorithm box with
pseudo-code, as well as the actual source code, are included in the supplementary.

**[R2] "msgs vs. no msgs"**: In no-msgs case, a limb can't get to know about other limbs it is not directly connected to,
while in msg passing, a limb can get to know so by learning to pass meaningful messages.

**[R2] "designs hand-crafted by the authors rather than design experts"**: We made our best effort to design a
working monolithic architecture. We tried linear morphology for standing and star-shaped for locomotion. A random
morphology doesn't work well for standing, but does okay for locomotion.

**[R2] "which axis is the surface height grid projected"**: It is the y-coordinate height of that block of the floor.

**[R3] train modular graph policy on fixed morphology**: Upon R3's great suggestion, we trained a modular DGN
policy for static morphology. The performance is significantly better than 'monolithic policy, static graph' but worse
than our final self-assembling DGN. This suggests that modularity is the key component for generalization. We do not
have the space to include plot here, but will include in the paper. Nevertheless, regardless of generalization properties,
one of the main contribution of our work is showing how could dynamic agents be trained to self-assemble.

**[R3] "From the video it seems that the robot sometimes breaks apart"**: Those are all due to the individual agent
producing the unlink action. The agent cannot break in other conditions in the current set up.

[Meta-Review · NeurIPS 2019]

The reviewers praise the paper's goal of integrating a model for dynamic morphologies and deep RL, as well as the clarity of its writing. Several reviewers bring up concerns around the issue of baselines. The reviewers ask for more comparisons with baselines in situations where both the proposed method and the baselines work, more empirical evidence about the reproducability of the proposed method's performance, and evidence for the strength of the baselines. Some of the reviewers' concerns were not addressed by the authors in their response. I strongly encourage the authors address these concerns in the camera ready version of the manuscript, either in the writing of the paper or with new experiments, or ideally with both. I look forward to reading the camera ready version of the manuscript with these considerations in place.